# ASO-LoRA: Attribution Scores-based Soft Orthogonality Low-Rank Adaptation for Large Language Model Continual Learning

## Abstract

Continual learning (CL) remains a critical challenge when applying large language models (LLMs) to real-world situations. On the one hand, billions of parameters for LLMs add a huge computing overhead to CL. Existing techniques, on the other hand, solely address catastrophic forgetting while ignoring the possibility of knowledge transfer between tasks. Facing these challenges, we propose Attribution Scores-based Soft Orthogonality Low-Rank Adaptation (ASO-LoRA), an effective and efficient framework that simultaneously facilitates knowledge transfer while mitigating catastrophic forgetting. Specifically, ASO-LoRA initially assigns task-specific parameter subspaces for new tasks utilizing multi-LoRA modules, enabling for efficient training and inference without relying on task labels. Then, ASO-LoRA leverages attribution scores to evaluate task similarity and suggests gradient steps in a soft orthogonal direction between task-specific subspaces, achieving a balance between knowledge transfer and preservation. Experiments are carried out on both the T5-large and the LLaMA2-7B, showing ASO-LoRA's suitability as a plug-in CL solution for general Transformer-based LLMs. Experimental results on CL benchmarks demonstrate that ASO-LoRA outperforms other strong baselines.

## 1 Introduction

Continuous learning (CL) is essential for applying language models in real-world scenarios, as presenting training data sequentially from varied distributions of different tasks can result in catastrophic forgetting (Wang et al., 2024b). Despite the remarkable performance of recently published large language models (LLMs) such as GPT-4 (Achiam et al., 2023b), Llama (Touvron et al., 2023), and DeepSeek (Liu et al., 2024a) across various tasks, CL still remains a significant challenge, as LLMs are not suited for frequent retraining due to the notable training costs related to their large scale (Wu et al., 2024).

In contrast to its employment in smaller models, efficient CL is of great importance for LLMs since implementing CL with billions of parameters incurs large computing costs. To realize efficient CL, recent research has made use of the parameter-efficient tuning frameworks for LLMs. Razdaibiedina et al. (2023) learns a new soft prompt for each task and concatenates it with previously learnt prompts, while freezing the base model. O-LoRA (Wang et al., 2023) proposes to incrementally learns new tasks in an orthogonal subspace while fixing the LoRA parameters learned from past tasks to minimize catastrophic forgetting. However, these methods tend to address only catastrophic forgetting, neglecting the possibility of transferring knowledge between tasks. SAPT (Zhao et al., 2024a) proposes to align the learning and selection of LoRA parameters via the shared attentive learning and selection module, addressing catastrophic forgetting and knowledge transfer simultaneously. Despite being PET-agnostic, SAPT introduces new modules to the architecture, increasing its complexity.

This research presents a simple yet efficient methodology for continual learning (CL) in large language models (LLMs), addressing both catastrophic forgetting and knowledge transfer. For the efficient manner, our study is based on Low-Rank Adaptation (LoRA) (Hu et al., 2022a), an efficient PET method that demonstrates how fine-tuning a specific low-rank subspace for new tasks can

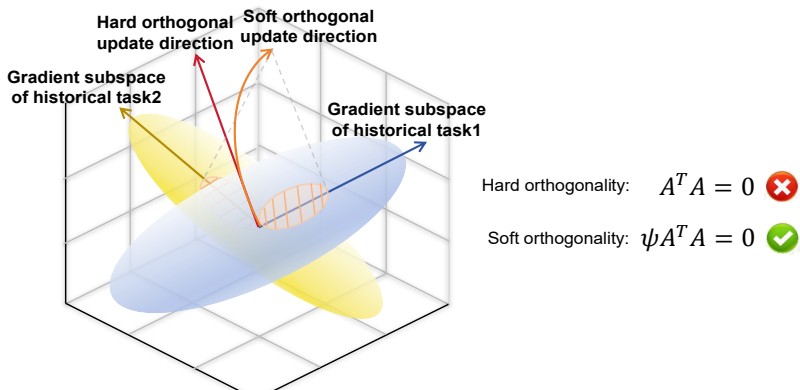

Figure 1: The key innovation lies in ASO-LoRA's dual capability to mitigate catastrophic forgetting of historical tasks while simultaneously facilitating inter-task knowledge transfer. ASO-LoRA updates the parameters subspace of new task through a soft orthogonal direction (orange curved arrow) with the historical tasks, instead of updating through a hard orthogonal direction (red straight arrow), to constrain the gradient updates of the current task to be softly orthogonal to the gradient subspace of the past tasks.

lead to competitive performance. Recent studies (Wang et al., 2023; Saha et al., 2021) suggest that optimizing along directions orthogonal to historical tasks' gradient subspaces can meet the purpose of mitigating catastrophic forgetting, by minimizing interference with their loss functions.

Inspired by these, we propose Attribution Scores-based Soft Orthogonality Low-Rank Adaptation (ASO-LoRA), an effective and efficient framework that facilitates transfer of knowledge while reducing catastrophic forgetting. We begin by assigning task-specific parameter subspaces for new tasks utilizing multi-LoRA modules, leaving the LLMs' parameters frozen. Multi-LoRA also enables inference without relying on task labels, allowing for generalization to previously unknown tasks. Then we improve the original method, which only takes gradient steps in the orthogonal direction, by hypothesizing that different tasks may share similar knowledge that could be transferred to enhance their task capabilities. As shown in Figure 1, we use attribution scores (Dai et al., 2021) to evaluate task similarity and propose taking gradient steps in a soft orthogonal direction between task-specific subspaces, achieving a balance between knowledge transfer and preservation.

We conduct experiments using the encoder-decoder-based T5-large and decoder-only-based LLaMA2-7B models, demonstrating ASO-LoRA as a plug-in CL solution for general Transformer-based LLMs. Experimental results on CL benchmarks show that ASO-LoRA outperforms other strong baselines.

## 2 RELATED WORKS

**Continual learning in LLMs** refers to the paradigm where LLMs sequentially acquire new knowledge from non-stationary data distributions while preserving learned capabilities (Wang et al., 2024b; Wu et al., 2024), and can be categorized into these types: Replay-based approach (Smith et al., 2024; Petit et al., 2023), Regularization-based approach (Zhao et al., 2024c), Optimization-based approach (Wang et al., 2022a), and Architecture-based approach (Han et al., 2025).

**Parameter-Efficient Tuning (PET)** adapts models by optimizing performance through updates to a minimal set of parameters (without direct modification of original parameters), employing Adapters (Wang et al., 2022b), soft prompts (Liu et al., 2024c), or low-rank adaptations (LoRA) (Liu et al., 2024b), to significantly reduce computational costs (Coleman et al., 2025).

Inspired by the above works, our method incorporates the LoRA mechanism while introducing soft orthogonality regularization to the loss function, achieving an optimal balance between model performance and training efficiency for LLMs in continual learning scenarios. Due to space constraints, please refer to the Appendix B for more detailed related works.

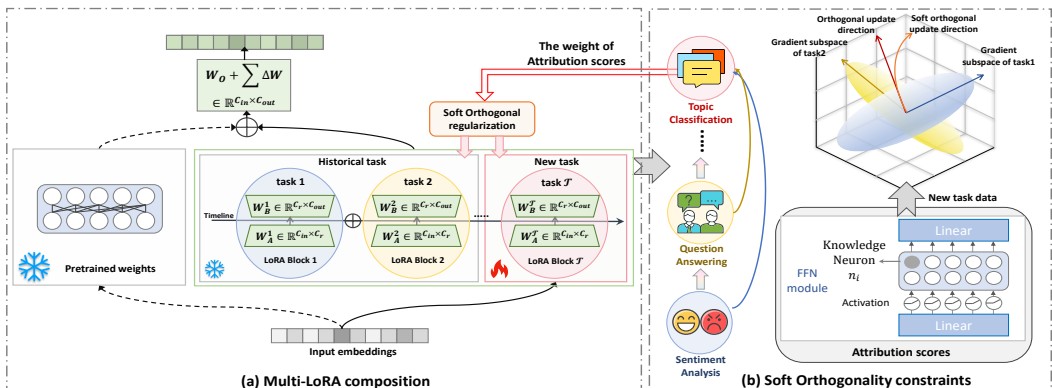

Figure 2: The overall framework of ASO-LoRA, employing multi-LoRA composition with a soft orthogonality constraint. a) We progressively train a new LoRA block for each incoming new task while freezing all historical LoRA blocks, and sequentially stack it upon existing LoRA blocks for the inference stage. b) The parameter updates for the new task's subspace adhere to a soft orthogonality constraint as a regularization term: we derive a soft orthogonality coefficient by quantifying the similarity between the attribution scores of the current and historical tasks, instead of enforcing the subspaces to be strictly orthogonal to mitigate catastrophic forgetting. This soft orthogonality allows for partial overlap in the tasks' subspaces, facilitating the cross-task knowledge transfer.

## 3 METHODOLOGY

Figure 2 presents an overview of the whole framework. ASO-LoRA utilizes a multi-LoRA composition mechanism with soft orthogonality constraints to enable continual learning of incremental tasks. Next, we will present a comprehensive exposition of our proposed method.

### 3.1 PRELIMINARY

**Task Definition**: Continual learning (CL) refers to a learning paradigm in which a model $M$, parameterized by $\theta$, sequentially acquires knowledge from a series of tasks' data $\{D_1, D_2, ..., D_{\mathcal{T}}\}$ over $\mathcal{T}$ learning stages. Each task $t$ consists of a set of input and target output pairs $\{(x_i^t, y_i^t)\}_{i=1}^{n}$. The objective of CL is to optimize the model's parameters $\theta$ such that $M$ averagely performs well on all learned tasks:

$$max_\theta \sum_{t=1}^{\mathcal{T}} \sum_{\{(x_i^t, y_i^t)\}_{i=1}^{n} \in D_t} log p_M(y_i^t | x_i^t) \tag{1}$$

**Continual learning using Multi-LoRA composition.** We adopt LoRA, an efficient PET method to fine-tune a specific low-rank subspace of trainable parameters for new tasks, while keeping the pretrained parameters of the LLMs frozen. LoRA does not rely on task IDs during inference, therefore keeping LLMs' generalization ability on unseen tasks. Specifically, LoRA freezes the LLMs' pretrained weight matrix $W_{base} \in \mathbb{R}^{d \times k}$ without receiving gradient updates to preserve the acquired knowledge. Trainable low-rank decomposition matrices $A_t \in \mathbb{R}^{d \times r}$ and $B_t \in \mathbb{R}^{r \times k}$ are introduced for each new task $t$ learning in parallel pathways, with $r \ll min(d, k)$.

To enable continual task adaptation for pretrained LLMs, we employ the multi-LoRA mechanism, in which a separate LoRA block $LoRA_t = \{A_t, B_t\}$ is trained for each downstream task $t$. The base LLM attached with $LoRA_t$ is defined as $M_{LoRA_t}$. The updated weight matrix $W_t$ of $M_{LoRA_t}$ for specific task $t$ is as follows:

$$W_t = W_{base} + \Delta W_t = W_{base} + A_t B_t \tag{2}$$

As the number of continually learned tasks increases, we can integrate the updated LoRA parameters into the base LLM parameters during inference, as shown in Eq.(3).

$$W_{merge_{\mathcal{T}}} = W_{base} + \sum_{t=1}^{\mathcal{T}} A_t B_t \tag{3}$$

## 3.2 Enhancing Multi-LoRA with Attribution scores based Soft Orthogonality

The vanilla CL with multi-LoRA, as described in Eq.(3), fails to account for the interactions between LoRA blocks, leading to instability in the fusion process and catastrophic forgetting. To address the issues raised above, we propose to enhance LoRA with Attribution Scores based Soft Orthogonality (ASO-LoRA), taking into account both catastrophic forgetting and knowledge transfer between tasks while integrating LoRA blocks. In particular, ASO-LoRA uses orthogonal regularization among the subspaces of multi-LoRA blocks to prevent catastrophic forgetting following previous studies (Wang et al., 2023). It further introduces attribution scores to softly adjust the orthogonal regularization, thereby improving positive inter-task knowledge transfer.

Formally, ASO-LoRA starts by learning new tasks in a direction orthogonal to the LoRA subspaces of historical tasks while freezing historical LoRA parameters. For each task $t$, $A_t = [a_t^1, a_t^2, ..., a_t^r]$ of $LoRA_t = \{A_t, B_t\}$ is approximated as the core of the task-related subspace, where each vector in $B_t = [b_t^1, b_t^2, ..., b_t^r]$ represents the linear weighting coefficients of the column vectors in $A_t$. Achieving orthogonality between the subspaces of the new task $\mathcal{T}$ and each historical task $t \in \{1, ..., \mathcal{T} - 1\}$ can be expressed as:

$$A_t^T A_\mathcal{T} = 0 \tag{4}$$

The hard orthogonality defined in Eq (4) simply considers the difference between tasks while ignoring the sharing of knowledge, resulting in inefficiency. We hypothesize that the relationship between LoRA blocks is not strictly orthogonal and that interactions between LoRA blocks may yield knowledge transfer with beneficial effects. To capture the shared knowledge, we further introduce the concept of a dynamic soft orthogonality coefficient $\psi_{t,\mathcal{T}}^{soft}$. $\psi_{t,\mathcal{T}}^{soft}$ decreases as more knowledge is shared between two LoRA blocks, indicating a stronger enhanced parametric correlation. In contrast, the coefficient increases as the LoRA blocks become more orthogonal, indicating a decrease in the relevance of the knowledge of tasks represented by the LoRA block. With the soft coefficient factor $\psi_{t,\mathcal{T}}^{soft}$, the hard orthogonality defined in Eq.(4) can be reformulated as the attribution score-based soft orthogonality $ASO_{t,\mathcal{T}}$:

$$ASO_{t,\mathcal{T}} = \psi_{t,\mathcal{T}}^{soft} A_t^T A_\mathcal{T} = 0 \tag{5}$$

The soft coefficient factor $\psi_{t,\mathcal{T}}^{soft}$ is calculated by the weight of attribution scores $w_{t,\mathcal{T}}^{attr}$ between the new task $\mathcal{T}$ and historical task $t(t < \mathcal{T})$ as below:

$$\psi_{t,\mathcal{T}}^{soft} = 1 - w_{t,\mathcal{T}}^{attr} \tag{6}$$

Finally, the CL training objective for the new task $\mathcal{T}$ is defined as:

$$\mathcal{L} = \sum_{(x,y) \in D_\mathcal{T}} log p_{M_{LoRA_\mathcal{T}}}(y|x) + \lambda \sum_{t=1}^{\mathcal{T}-1} \mathcal{L}_{ASO}(A_t, A_\mathcal{T})$$
$$\mathcal{L}_{ASO}(A_t, A_\mathcal{T}) = \sum_{j,k} ||ASO_{t,\mathcal{T}}[j,k]||^2 \tag{7}$$

Where $ASO_{t,\mathcal{T}}$ is the attribution score-based soft orthogonality defined in Eq.(5). $ASO_{t,\mathcal{T}}[j,k]$ denotes the element at the $j-th$ row and $k-th$ column of $ASO_{t,\mathcal{T}}$, and $\lambda$ is the weight of the soft orthogonality loss.

Next, we present the details about how to derive the weight of attribution scores $w_{t,\mathcal{T}}^{attr}$.

## 3.3 The weight of Attribution scores

Before introducing the weight $w_{t,\mathcal{T}}^{attr}$ of the attribution score, we first introduce the concept of Knowledge Neuron.

**Knowledge Neurons (KNs)** are neurons in the Transformer's Feed-forward Network (FFN) memories that store factual knowledge, discovered by recent studies (Dai et al., 2021). FFN is responsible for applying a non-linear transformation to the hidden state $H$, given by the multi-head attention (MHA) module of Transformer:

$$FFN(H) = GELU(HW^{l^1})W^{l^2} = Neurons W^{l^2} \tag{8}$$

$W^{l^1}$ and $W^{l^2}$ are weight parameter matrices of the FFN layers. For simplicity, we omit the scaling factor in MHA and the bias terms in FFN.

**Attribution scores** evaluate the contribution of each knowledge neuron in the LLM to the knowledge expression of data (Dai et al., 2021) based on the integrated gradients. We utilize attribution scores to assess how similar the model with the new-task LoRA is to models with historical-task LoRA blocks, allowing us to establish the proper level of orthogonal constraints between the LoRA blocks.

Given an input data $x_\mathcal{T}$ of new task $\mathcal{T}$, we first define the output $P^\mathcal{T}_{M_{LoRA_t}}(\hat{n}_t^i)$ as the probability of the correct answer predicted by $M_{LoRA_t}$ (base model attached to a LoRA of each specific task $t$):

$$P^\mathcal{T}_{M_{LoRA_t}}(\hat{n}_t^i) = p_{M_{LoRA_t}}(y^*|x_\mathcal{T}, n_t^i = \hat{n}_t^i) \tag{9}$$

where $y^*$ denotes the correct answer; $n_t^i$ denotes the knowledge neuron of $M_{LoRA_t}$ to be calculated, and $\hat{n}_t^i$ is a given constant that $n_t^i$ is assigned to.

To determine the attribution score $Attr(n_t^i)$, we gradually alter $n_t^i$ from 0 to its original value $\bar{n}_t^i$ calculated by the $M_{LoRA_t}$, while simultaneously integrating the gradients:

$$Attr(n_t^i) = \bar{n}_t^i \int_{\sigma=0}^1 \frac{\delta P^\mathcal{T}_{M_{LoRA_t}}(\sigma\bar{n}_t^i)}{\delta n_t^i} d\sigma \tag{10}$$

$\frac{P^\mathcal{T}_{M_{LoRA_t}}(\sigma\bar{n}_t^i)}{\delta n_t^i}$ calculates the gradient of the model output with regard to $n_t^i$. As $\sigma$ varies from 0 to 1, $Attr(n_t^i)$ accumulates the cumulative impact of modifying $n_t^i$ on the change of output probability.

Since it is difficult to directly calculate continuous integrals, the Riemann approximation (Roe, 1981) method is used to estimate the value of integrated gradients, and we set the number of approximation steps $m = 20$:

$$Attr(n_t^i) = \frac{\bar{n}_t^i}{m} \sum_{k=1}^m \frac{\delta P^\mathcal{T}_{M_{LoRA_t}}(\frac{k}{m}\bar{n}_t^i)}{\delta n_t^i} \tag{11}$$

When a knowledge neuron $n_t^i$ significantly impacts the expression of knowledge, the resulting gradient becomes prominent, leading to high integration values $Attr(n_t^i)$.

For $M_{LoRA_t}$, we assemble the attribution scores corresponding to all knowledge neurons $[n_t^1, ..., n_t^N]$, denoting them as a vector:

$$Attr\_vec_{M_{LoRA_t}} = [Attr(n_t^1), ..., Attr(n_t^N)] \tag{12}$$

Where $N$ represents the number of $M_{LoRA_t}$'s KNs.

Using the aforementioned, we can calculate the weight of attribution scores $w_{t,\mathcal{T}}^{attr}$ as shown below. We quantify the influence of each historical task $t(t < \mathcal{T})$ on the current task $\mathcal{T}$ by computing the similarity between their attribution vectors for the current task $\mathcal{T}$'s data:

$$w_{t,\mathcal{T}}^{attr} = Sim(Attr\_vec_{M_{LoRA_t}}, Attr\_vec_{M_{LoRA_\mathcal{T}}}) \tag{13}$$

$w_{t,\mathcal{T}}^{attr}$ evaluates the transferability of knowledge from previous tasks $\{1, 2, ..., \mathcal{T}-1\}$ to new task $\mathcal{T}$. A higher $w_{t,\mathcal{T}}^{attr}$ indicates greater knowledge sharing between tasks $t$ and $\mathcal{T}$, suggesting that $LoRA_t$ is more likely to positively influence $LoRA_\mathcal{T}$ block.

# 4 EXPERIMENTS

## 4.1 EXPERIMENTAL SETUPS

### 4.1.1 DATASETS

**CL benchmarks**: Following Wang et al. (2023)'s work, we conduct comprehensive evaluations of ASO-LoRA against state-of-the-art baselines on four standard continual learning benchmarks for language models: AG News, Amazon Reviews, DBpedia, and Yahoo Answers (Zhang et al., 2015).

**Longer benchmarks**: To further validate our approach, we conduct extensive experiments on longer task sequences, incorporating 15 common tasks used for language models (Razdaibiedina et al., 2023), including the AG News, Amazon Reviews, DBpedia, Yahoo Answers, and Yelp reviews from standard CL benchmarks; MNLI, QQP, RTE, SST2 from GLUE benchmark (Wang et al., 2018); WiC, CB, COPA, MultiRC, BoolQ from SuperGLUE (Wang et al., 2019), and the IMDB movie reviews (Maas et al., 2011).

We explore 6 different orders of the benchmarks to validate the methods' efficacy across diverse continual learning scenarios.

### 4.1.2 METRICS

Following Wang et al. (2024a), we use the Average Accuracy (AA) to evaluate the performance of ASO-LoRA in the CL scenario, which is the average accuracy of all tasks after the model finishes training on the latest task $\mathcal{T}$:

$$AA = \frac{1}{\mathcal{T}} \sum_{i=1}^{\mathcal{T}} a_{\mathcal{T},i} \tag{14}$$

We also employ forgetting measure (FM) and forward transfer (FWT) as evaluation metrics to comprehensively evaluate our approach.

### 4.1.3 BASELINES

Based on Wang et al. (2023)'s work, we evaluate our method against strong competitive baselines: (1) **Non-CL baselines**: SeqFT(de Masson D'Autume et al., 2019) and PerTaskFT. PerTaskFT is considered as the upper bound. (2) **LoRA-based**: SeqLoRA, IncLoRA, MoELoRA(Luo et al., 2024), MoCL(Wang et al., 2024c) and O-LoRA(Wang et al., 2023). (3) **Traditional CL baselines**: Replay(Chaudhry et al., 2019), EWC(Kirkpatrick et al., 2017), LwF(Li & Hoiem, 2017), L2P(Wang et al., 2022c), and LFPT5(Qin & Joty, 2021). See Appendix C.2 for details of these baselines.

### 4.1.4 IMPLEMENTATION DETAILS

ASO-LoRA employs the generalization-friendly instruction tuning as the training paradigm, capturing the underlying commonalities of tasks. We implement ASO-LoRA on two representative Transformer architectures: the encoder-decoder model T5-large (710M) (Raffel et al., 2020) and the decoder-only model LLaMA2-7B (Touvron et al., 2023), highlighting ASO-LoRA's applicability as a plug-in continual learning solution for general Transformer-based language models. The similarity in Eq.(13) employs Spearman's rank correlation coefficient. We train the models with one epoch, using the AdamW (Loshchilov & Hutter, 2017) optimizer in a batch size of 64 with learning rate $1 \times 10^{-3}$ for each experiment. The weight decay is 0 while the dropout rate is set as 0.1. The weights of the soft orthogonality loss follow Wang et al. (2023)'s work. Results are reported as the average of 3 runs. Our setup consists of a four-core CPU and eight NVIDIA Tesla A100 GPUs.

For more details on task orders, task details, metrics, and baselines, refer to the Appendix C.

## 4.2 MAIN RESULTS AND ANALYSIS

Table 1 displays a comprehensive performance comparison between ASO-LoRA and baselines on both CL benchmarks and extended longer benchmarks. We evaluate the effectiveness of ASO-LoRA for CL scenarios from three perspectives:

Table 1: The main averaged accuracy (AA) results on two series of benchmarks with the T5-large model (T5-710M), after training on the last task. The results of CL baselines are referred from Wang et al. (2023)'s work, while the results of MoCL and MoELoRA are from Du et al. (2024).

| | CL benchmarks | | | | Longer benchmarks | | | |
| --- | --- | --- | --- | --- | --- | --- | --- | --- |
| | Order-1 | Order-2 | Order-3 | Avg. | Order-4 | Order-5 | Order-6 | Avg. |
| SeqFT | 18.9 | 24.9 | 41.7 | 28.5 | 7.4 | 7.4 | 7.5 | 7.4 |
| PerTaskFT | 72.5 | 72.5 | 72.5 | 72.5 | 78.0 | 78.0 | 78.0 | 78.0 |
| Replay | 55.2 | 56.9 | 61.3 | 57.8 | 55.0 | 54.6 | 53.1 | 54.2 |
| EWC | 48.7 | 47.7 | 54.5 | 50.3 | 45.3 | 44.5 | 45.6 | 45.1 |
| LwF | 54.4 | 53.1 | 49.6 | 52.3 | 50.1 | 43.1 | 47.4 | 46.9 |
| L2P | 60.3 | 61.7 | 61.1 | 60.7 | 57.5 | 53.8 | 56.9 | 56.1 |
| LFPT5 | 67.6 | 72.6 | 77.9 | 72.7 | 70.4 | 68.2 | 69.1 | 69.2 |
| SeqLoRA | 62.6 | 61.5 | 69.3 | 64.5 | 56.7 | 52.7 | 15.9 | 41.8 |
| IncLoRA | 69.5 | 64.9 | 70.9 | 68.4 | 59.0 | 62.6 | 62.3 | 61.3 |
| MoCL | 75.6 | 75.4 | 76.7 | 75.9 | - | - | - | - |
| MoELoRA | 52.8 | 49.6 | 59.8 | 54.1 | 36.3 | 31.4 | 15.1 | 27.6 |
| O-LoRA | 75.6 | **78.1** | 72.1 | 75.3 | 71.6 | **69.3** | 75.8 | 72.2 |
| **ASO-LoRA** | **76.2** | 77.5 | **77.3** | **77.0** | **73.7** | 67.2 | **77.4** | **72.8** |

**Performance on CL benchmarks**: As evidenced by the results, ASO-LoRA consistently outperforms all baselines across different task orders. Notably, ASO-LoRA achieves significant improvements of 4.5% and 1.7% compared to PerTaskFT and O-LoRA, respectively. These demonstrate that ASO-LoRA effectively mitigates catastrophic forgetting in continual learning scenarios while successfully leveraging previously acquired knowledge. Notably, unlike conventional CL baselines, ASO-LoRA requires neither full-parameter training nor historical data storage, thereby achieving significant computational efficiency while preserving knowledge from the pertaining stage.

**Performance on longer general benchmarks**: Following Wang et al. (2023), we assess ASO-LoRA on a more challenging scenario, involving sequential training across 15 extended tasks. As illustrated in Table 1, ASO-LoRA achieves superior performance compared to almost all baselines in addressing longer continual learning problems, demonstrating ASO-LoRA's robust adaptability to more complex scenarios. However, PerTaskFT maintains higher performance, indicating that sustained knowledge preservation in long task sequences remains an open challenge.

**Results on other Transformer-based structures**: To further validate the generalizability of ASO-LoRA across Transformer-based architectures, we extend our experiments on the decoder-only LLaMA-7B model using standard CL benchmarks. As evidenced in Table 2, ASO-LoRA achieves a leading performance, with an average improvement of 0.9% over O-LoRA. These empirical evidences confirm ASO-

Table 2: The main AA results on CL benchmarks with the Llama-7B model, after training on the last task.

| Model | CL benchmarks | | | |
| --- | --- | --- | --- | --- |
| | Order-1 | Order-2 | Order-3 | Avg. |
| SeqLoRA | 77.2 | 76.6 | 78.0 | 77.2 |
| IncLoRA | 32.7 | 49.1 | 35.5 | 39.1 |
| O-LoRA | 73.7 | 75.0 | 76.5 | 75.1 |
| **ASO-LoRA** | **75.0** | 74.8 | **78.1** | **76.0** |

LoRA's plug-and-play adaptability, suggesting its broad applicability across diverse Transformer-based models for complex continual learning scenarios. We include the results of the LLaMA architecture on longer benchmarks in the Appendix D.1.

## 4.3 THE IMPACT OF SOFT ORTHOGONALITY

To facilitate a more intuitive analysis of Soft Orthogonality's effectiveness, we further display the comparative task-specific results after complete model training across three distinct task orderings (Order1-Order3) on both T5 and Llama structures, as listed in Table 3 and 8.

Table 3: Results on individual tasks after completing training on the final task with T5-large across Order 1&2&3. The left-to-right ordering of benchmarks corresponds to the task training order.

| Model | | CL benchmarks per-task results | | | |
|---|---|---|---|---|---|
| | Sequences | Dbpedia | Amazon | Yahoo | Ag |
| | PerTaskFT | 97.6 | 34.7 | 70.0 | 87.5 |
| Order1 | IncLoRA | 81.3 | 38.6 | 68.0 | 89.9 |
| | O-LoRA | 89.6 | 56.4 | 67.5 | 88.8 |
| | **ASO-LoRA** | 88.1 | **57.9** | **71.1** | 87.5 |
| | Sequences | Dbpedia | Amazon | Ag | Yahoo |
| | PerTaskFT | 97.6 | 34.7 | 87.5 | 70.0 |
| Order2 | IncLoRA | 72.0 | 44.2 | 70.1 | 73.5 |
| | O-LoRA | 96.8 | 56.4 | 87.7 | 71.8 |
| | **ASO-LoRA** | 91.3 | **58.5** | **87.9** | **72.1** |
| | Sequences | Yahoo | Amazon | Ag | Dbpedia |
| | PerTaskFT | 70.0 | 34.7 | 87.5 | 97.6 |
| Order3 | IncLoRA | 65.1 | 44.9 | 75.0 | 98.4 |
| | O-LoRA | 69.7 | 33.3 | 86.6 | 98.7 |
| | **ASO-LoRA** | **70.3** | **55.1** | 85.2 | **98.9** |

Regardless of task ordering, ASO-LoRA demonstrates superior performance on most tasks compared to O-LoRA and the fine-tuning-only upper bound. Although ASO-LoRA exhibits a lower AA score than O-LoRA on Order2, it achieves greater performance improvements across more individual tasks. Preliminary analysis suggests that potential negative backward transfer from later tasks to the initial task may lead to performance degradation. For Order3, ASO-LoRA achieves a significant 21.8% improvement over O-LoRA on task Amazon. These evidences still prove that the proposed Soft Orthogonality mechanism extends the interaction subspaces between LoRA blocks within PET frameworks, relaxing hard orthogonal constraints while preserving low-rank adaptation benefits. Refer to Appendix D.2 for the results and analysis of Llama structure.

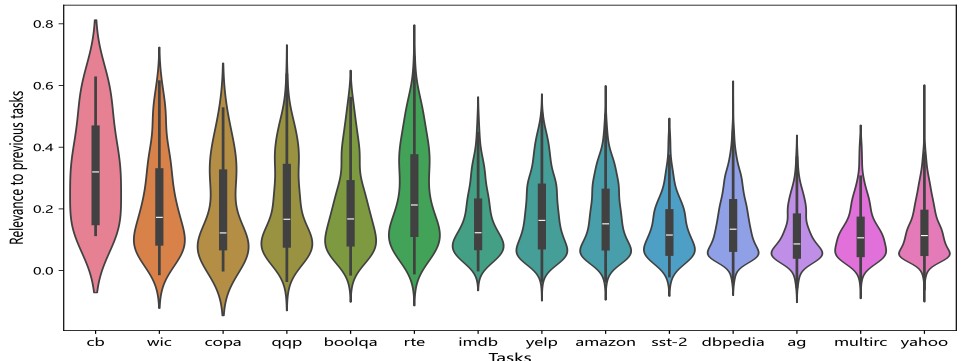

Figure 3: Longer benchmarks' Violin Plot of $w^{attr}$, displaying the distribution of similarity coefficient $w^{attr}$ across different tasks during the training stages. Central white dot represents the median or mean of the data, while a wider section indicates higher data density (more points near that value).

### 4.4 THE RELATIONSHIPS AMONG TASKS ON LONGER BENCHMARKS

We employ the violin plot to visualize the similarity coefficient $w^{attr}$ between the new task and historical tasks, as proposed in Eq.(13), analyzing their inter-task correlations and mutual influences.

Figure 3 presents the cross-task correlation coefficients between new and historical tasks under Order4. Task Cb exhibits a strong correlation with the preceding task MNLI. Since both tasks belong to the Natural Language Inference (NLI) category within the GLUE benchmark, this finding aligns with our hypothesis. Likewise, the high similarity between the tasks of RTE and BoolQA can be attributed to their common source from Wikipedia.

### 4.5 THE EFFECT ON CATASTROPHIC FORGETTING AND KNOWLEDGE TRANSFER

To comprehensively evaluate Soft Orthogonality's efficacy in mitigating catastrophic forgetting and facilitating knowledge transfer, we introduce additional metrics: FM and FWT. These metrics enable systematic comparison with the mere orthogonal-constrained strong baseline O-LoRA. Figure 4 reveals that ASO-LoRA obviously achieves superior knowledge transfer over O-LoRA by more effectively leveraging prior task knowledge on all structures. As the tasks increase, ASO-LoRA exhibits progressively reduced catastrophic forgetting, ultimately outperforming O-LoRA in long-term knowledge preservation on T5 structure. These substantiate our hypothesis that soft orthogonality offers a principled solution for balancing knowledge preservation and transfer in CL.

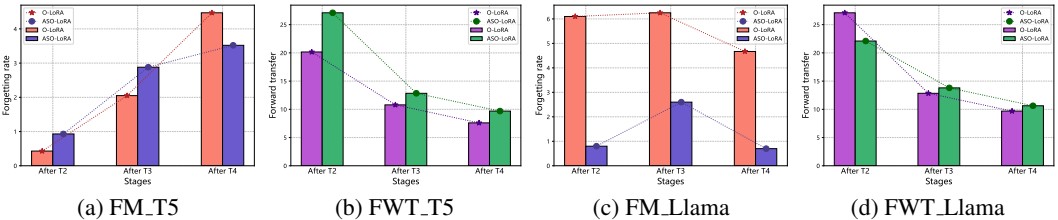

|       (a) FM_T5       |       (b) FWT_T5       |       (c) FM_Llama       |       (d) FWT_Llama       |

Figure 4: ASO-LoRA vs. O-LoRA on standard CL benchmarks using metrics FM and FWT. Lower FM values indicate stronger resistance to catastrophic forgetting and better knowledge retention capabilities, while higher FWT values demonstrate more effective utilization of prior knowledge and superior knowledge transfer.

### 4.6 THE OUTPUT SIMILARITY OF DIFFERENT TASKS ON LONGER BENCHMARKS

To investigate how our method enhances model performance distinct from O-LoRA, we visualize the variation in product magnitudes between O-LoRA and ASO-LoRA augmented by soft orthogonal in Figure 5. We employ heatmaps to depict the similarity of final output distributions across different tasks. Compared to O-LoRA, ASO-LoRA achieves higher task similarity, thereby strengthening the inter-task connections. This difference, driven by the distinct interaction patterns between LoRA blocks, indicates that our method facilitates more effective knowledge transfer across tasks, which in turn contributes to improved performance in continual learning scenarios.

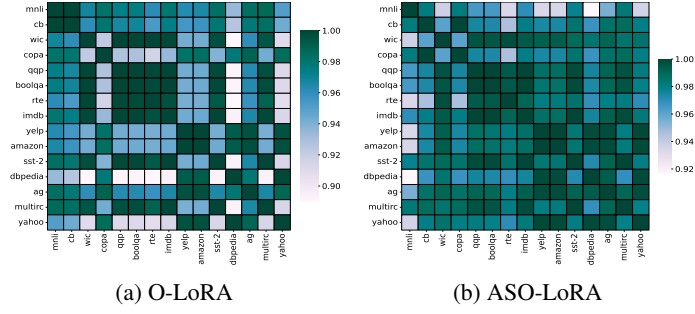

|          (a) O-LoRA          |          (b) ASO-LoRA          |

Figure 5: ASO-LoRA vs. O-LoRA on the final output distribution similarity across different tasks in the sequence of Order4 on T5-Large structure.

## 5 CONCLUSION

In this work, we present ASO-LoRA, an innovative continual learning framework that incorporates Attribution Score-based Soft Orthogonality for parameter-efficient adaptation. Experimental results demonstrate that ASO-LoRA outperforms strong baselines, effectively mitigating catastrophic forgetting while facilitating robust knowledge transfer across sequential tasks.

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

## A  THE USE OF LARGE LANGUAGE MODELS (LLMs)

We appreciate the assistance provided by GPT-4 (Achiam et al., 2023a) in writing aid and sentence-level polishing.

## B  RELATED WORKS

### B.1  CONTINUAL LEARNING IN LLMs

Continual learning in LLMs refers to the paradigm where LLMs sequentially acquire new knowledge from non-stationary data distributions while preserving learned capabilities (Wang et al., 2024b; Wu et al., 2024), and can be categorized into these types: **Replay-based approach** approximates and recovers old knowledge by storing old training samples (Chaudhry et al., 2019; Smith et al., 2024), or extracting information from prior feature representations (Zhu et al., 2022; Petit et al., 2023). **Regularization-based approach** adds explicit regularization terms to balance the old and new tasks, usually requiring storing a frozen copy of the old model (Kirkpatrick et al., 2017; Zhao et al., 2024c). **Optimization-based approach** explicitly designs the optimization programs to achieve continual learning (Wang et al., 2022a). **Architecture-based approach** investigates dynamic model

expansion (Yao et al., 2023; Han et al., 2025; Qin et al., 2022), combined with parameter isolation mechanisms (Wang et al., 2023; Zhao et al., 2024b) to minimize task interference in sequential learning scenarios.

### B.2 PARAMETER-EFFICIENT TUNING (PET)

PET has emerged as a resource-efficient approach for model adaptation (Coleman et al., 2025), aiming to optimize performance while updating only a small number of trainable parameters to reduce computational costs. Some works train adapters for the downstream task while keeping the pre-trained model parameters frozen (He et al., 2023; Wang et al., 2022b). Another line of research explores the integration of trainable tensors, called soft prompts, into model input representations (Lester et al., 2021; Liu et al., 2024c). Recently, reparameterization methods such as low-rank adaptations (LoRA) have garnered significant attention (Hu et al., 2022b; Dettmers et al., 2023; Liu et al., 2024b). These approaches avoid direct modification of original weight matrices, instead updating parameters through transformation functions operating on smaller parameter sets.

## C EXPERIEMTAL SETUPS

### C.1 METRCIS

Forgetting measure(FM) is also applied to calculate the memory stability of models. The forgetting of a task is calculated by the difference between its maximum performance obtained in the past and its current performance:

$$f_{j,} = \max_{i \in \{1, \dots, t-1\}} (a_{i,j} - a_{t,j}), \forall j < t \tag{15}$$

FM at the t-th task is the average forgetting of all old tasks:

$$FM_t = \frac{1}{t-1} \sum_{j=1}^{t-1} f_{j,t} \tag{16}$$

FWT evaluates the average influence of all old tasks on the current t-th task:

$$FWT_t = \frac{1}{t-1} \sum_{j=2}^{t} (a_{j,j} - \tilde{a}_j) \tag{17}$$

where $\tilde{a}_j$ is the accuracy of a base model trained with $D_j$ for the j-th task.

### C.2 BASELINES

We conduct comprehensive comparisons between our method and 10 baseline models, whose introductions are detailed as follows:

**SeqFT** (de Masson D'Autume et al., 2019): trains all model parameters sequentially across tasks without employing any regularization or replay techniques.

**PerTaskFT**: trains a separate model for each task.

**Replay** (Chaudhry et al., 2019): is fine-tuned on all parameters with a memory buffer mechanism, replaying stored prior samples to prevent knowledge forgetting.

**EWC** (Kirkpatrick et al., 2017): performs full-model fine-tuning with regularization constraints designed to preserve parameters critical for previously learned tasks.

**LwF** (Li & Hoiem, 2017): constrains the shared representation layer to be similar to its original state before learning new tasks.

**L2P** (Wang et al., 2022c): dynamically selects and updates prompts from the pool in an instance-wise manner based on input characteristics.

**LFPT5** (Qin & Joty, 2021): implements continuous soft prompt training for direct task solution and training sample generation.

**SeqLoRA**: trains the fixed-size LoRA parameters on a sequence of tasks without any regularization or replaying techniques.

**IncLoRA**: sequentially acquires a series of new tasks through incremental LoRA parameter expansion, without any regularization or replaying techniques.

**MOCL** (Wang et al., 2024c): continually adds new trainable PEFT parameters (LoRA) to language models and composes them with existing modules.

**MoELoRA** (Luo et al., 2024): considers LoRA as a Mixture of Experts, harnessing the collective modeling capacity of multiple experts to handle different domains while retaining LoRA's parameter-efficient characteristics.

**O-LoRA** (Wang et al., 2023): learns new tasks in different vector subspaces (low-rank) that are kept orthogonal to each other to prevent catastrophic forgetting.

### C.3 TASK DETAILS

We list the sequences of tasks used in our experiments in Table 4, while Table 5 provides representative task instruction templates.

Table 4: Six different orders of task sequences used for continual learning experiments. These orders are following Wang et al. (2023) and Razdaibiedina et al. (2023)'s works.

| Order | Task Sequence |
|---|---|
| 1 | dbpedia→amazon→yahoo→ag |
| 2 | dbpedia→amazon→ag→yahoo |
| 3 | yahoo→amazon→ag→dbpedia |
| 4 | mnli → cb → wic → copa → qqp → boolqa → rte → imdb → yelp → amazon → sst-2 → dbpedia → ag → multirc → yahoo |
| 5 | multirc → boolqa → wic → mnli → cb → copa → qqp → rte → imdb → sst-2 → dbpedia → ag → yelp → amazon → yahoo |
| 6 | yelp → amazon → mnli → cb → copa → qqp → rte → imdb → sst-2 → dbpedia → ag → yahoo → multirc → boolqa → wic |

Table 5: Instructions for different tasks

| Task | Prompt |
|---|---|
| NLI | What is the logical relationship between the "sentence 1" and the "sentence 2"? Choose one from the options. |
| QQP | Whether the "first sentence" and the "second sentence" have the same meaning? Choose one from the options. |
| SC | What is the sentiment of the following paragraph? Choose one from the options. |
| TC | What is the topic of the following paragraph? Choose one from the options. |
| BoolQA | According to the following passage, is the question true or false? Choose one from the options. |
| MultiRC | According to the following passage and question, is the candidate answer true or false? Choose one from the options. |
| WiC | Given a word and two sentences, whether the word is used with the same sense in both sentences? Choose one from the options. |
| COPA | Given a prompt sentence, a question, and two possible answers, which option is more reasonable to answer the question. Choose one from two options. |

## C.4 Comparison with CL methods

In Table 6, we compare ASO-LoRA with common CL methods. Our approach shows four distinct advantages: rehearsal-free, parameter-efficient, task-id-available, and knowledge-transferable.

Table 6: The comparison between ASO-LoRA and other continual learning methods. Specifically, RF indicates whether the method is rehearsal-free. PE indicates whether the method is parameter efficient. TIF indicates whether task identify is available during inference. KT indicates whether the method enables knowledge transfer.

| Method | RF | PE | TIF | KT |
|---|---|---|---|---|
| EWC (Kirkpatrick et al., 2017) | ✓ | | ✓ | ✓ |
| LwF (Li & Hoiem, 2017) | ✓ | | | ✓ |
| L2P (Wang et al., 2022c) | ✓ | ✓ | ✓ | |
| LFPT5 (Qin & Joty, 2021) | | ✓ | ✓ | ✓ |
| MoELoRA (Luo et al., 2024) | ✓ | ✓ | | |
| O-LoRA (Wang et al., 2023) | ✓ | ✓ | ✓ | |
| **ASO-LoRA** | ✓ | ✓ | ✓ | ✓ |

# D Supplementary Experiments

## D.1 Performance on other Transformer-based structures

To further validate the generalizability of ASO-LoRA across Transformer-based architectures, we extend our experiments on the decoder-only LLaMA-7B model using Longer CL benchmarks. As evidenced in Table 7, ASO-LoRA achieves significantly superior performance, with an average accuracy improvement of 5.4% compared to O-LoRA. These experimental results further confirm the plug-and-play adaptability of ASO-LoRA, demonstrating its broad applicability across various Transformer-based models in more complex continual learning scenarios.

Table 7: The main AA results on Longer benchmarks with the Llama-7B model, after training on the last task.

| Model | Longer benchmarks | | | |
|---|---|---|---|---|
| | Order-4 | Order-5 | Order-6 | Avg. |
| SeqLoRA | 0 | 11.9 | 9.0 | 6.9 |
| IncLoRA | 26.8 | 28.3 | 36.2 | 30.4 |
| O-LoRA | 46.4 | 61.9 | 58.1 | 55.5 |
| **ASO-LoRA** | **59.1** | **64.5** | **59.2** | **60.9** |

## D.2 The impact of Soft Orthogonality on Llama structure

We further analyze the task-specific performance of the final model on the LLaMA architecture after completing all training tasks, as detailed in Table 8. ASO-LoRA outperforms other baselines on almost all individual tasks, proving that our soft orthogonality can generally leverage the inter-task knowledge and potential knowledge transfer on broader structures.

# E Visualization

## E.1 The relationships among tasks on Longer benchmarks

We employ the violin plot to visualize the similarity coefficient $w^{attr}$ between the new task and historical tasks, as proposed in Eq.(13), analyzing their inter-task correlations and mutual influences.

Table 8: The results on individual tasks after completing training on the final task with Llama, across Order 1, Order 2, and Order 3. The left-to-right ordering of benchmarks corresponds to the task training order.

| Model | | CL benchmarks per-task results | | | |
|---|---|---|---|---|---|
| | Sequences | Dbpedia | Amazon | Yahoo | Ag |
| Order1 | IncLoRA | 8.7 | 0.0 | 31.4 | 90.7 |
| | O-LoRA | 96.3 | 41.6 | 66.1 | 90.9 |
| | **ASO-LoRA** | **98.3** | **56.3** | 65.7 | 80.0 |
| | Sequences | Dbpedia | Amazon | Ag | Yahoo |
| Order2 | IncLoRA | 52.2 | 23.5 | 51.5 | 69.4 |
| | O-LoRA | 97.8 | 45.4 | 89.0 | 67.9 |
| | **ASO-LoRA** | **97.9** | **51.3** | 78.9 | **70.6** |
| | Sequences | Yahoo | Amazon | Ag | Dbpedia |
| Order3 | IncLoRA | 4.7 | 26.2 | 12.6 | 98.5 |
| | O-LoRA | 64.0 | 61.1 | 82.4 | 98.6 |
| | **ASO-LoRA** | **69.4** | 58.2 | **86.0** | **98.9** |

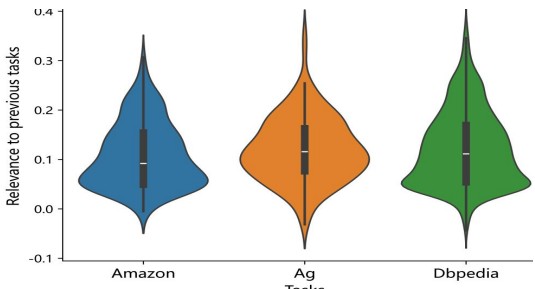

Figure 6: The Violin Plot of $w^{attr}$ on T5-large, displaying the distribution of similarity coefficient $w^{attr}$ across different tasks during the training stages. Central white dot represents the median or mean of the data, while a wider section indicates higher data density (more points near that value).

As illustrated in Figure 6, the higher similarity coefficient between Task Ag News and historical tasks indicates a positive influence of Ag News on earlier tasks. As evidenced in Table 3, ASO-LoRA obviously achieves a 21.8% performance improvement on Task Ag's previous task Amazon compared to O-LoRA. These results demonstrate that Soft Orthogonality productively leverages previously acquired task knowledge and effectively facilitates knowledge transfer.

### E.2 THE TREND OF PERFORMANCE IN THE CONTINUAL LEARNING STAGES

**Standard CL benchmarks**: As shown in Figure 7, we compare Average Accuracy trajectories of ASO-LoRA and O-LoRA during CL stages on standard CL benchmarks. Throughout the whole CL stages, ASO-LoRA exhibits consistent performance and generally outperforms O-LoRA in overall results. This further demonstrates that soft orthogonality between LoRA blocks corresponding to different tasks generally enhances the expression of related knowledge, while also achieving a favorable balance between mitigating catastrophic forgetting and facilitating knowledge transfer.

**Longer benchmarks**: Figure 8 further dipicts the Average Accuracy trajectories of ASO-LoRA and O-LoRA during CL stages on longer benchmarks. ASO-LoRA demonstrates greater overall stability and outperforms O-LoRA during longer continual learning phases. On Order4, although ASO-LoRA initially lags behind O-LoRA, its ability to mitigate catastrophic forgetting becomes increasingly evident as the number of tasks grows, eventually surpassing O-LoRA. On Order5, while ASO-LoRA does not ultimately outperform O-LoRA, it exhibits a more balanced performance and avoids the sharp performance drop observed in O-LoRA at epoch 8. On Order6, the two curves fol-

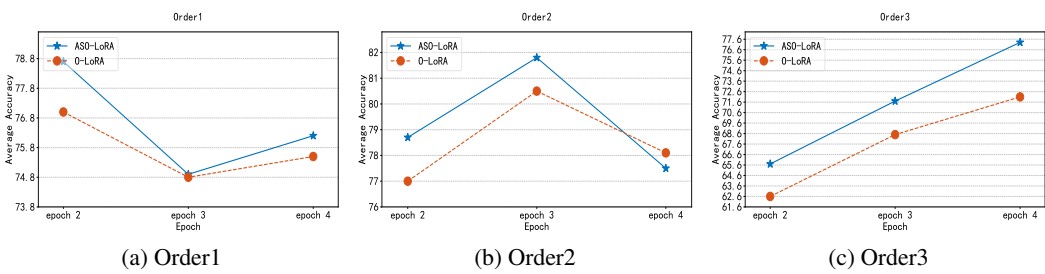

Figure 7: ASO-LoRA vs. O-LoRA on Average Accuracy (AA) during the CL stage on CL standard benchmarks (T5 structure), as new tasks arrive continuously. Each arrival of a new task corresponds to one epoch of training. The blue line represents the trend of ASO-LoRA'AA as tasks increment, while the red line corresponds to that of O-LoRA.

low a broadly aligned trajectory, yet one decreases as the other increases. ASO-LoRA consistently maintains an overall advantage.

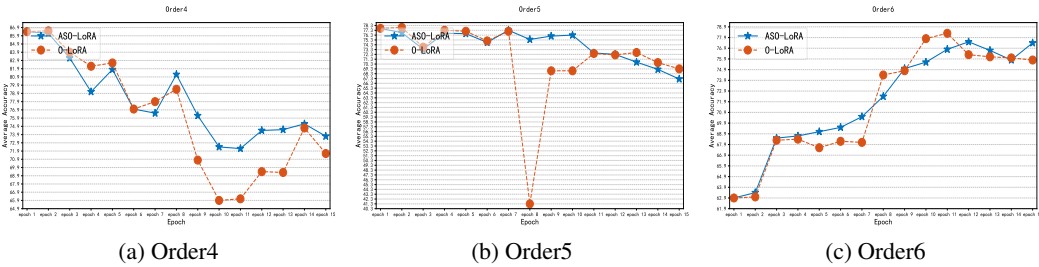

Figure 8: ASO-LoRA vs. O-LoRA on Average Accuracy (AA) during the CL stage on CL longer benchmarks, as new tasks arrive continuously. Each arrival of a new task corresponds to one epoch of training. The blue line represents the trend of ASO-LoRA'AA as tasks increment, while the red line corresponds to that of O-LoRA

We also investigate the training loss trajectories of ASO-LoRA and O-LoRA during CL stages on longer benchmarks, depicted in Figure 9. While following the same overall trend, ASO-LoRA exhibits lower and more stable training loss than O-LoRA, further reinforcing the suitability of ASO-LoRA for continual learning scenarios.

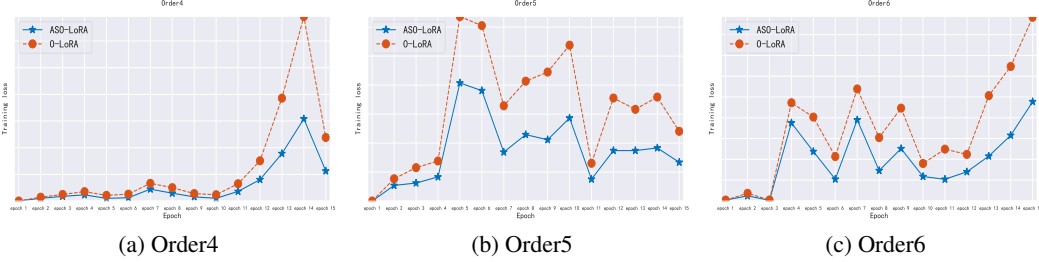

Figure 9: ASO-LoRA vs. O-LoRA on training loss during the CL stage on CL longer benchmarks, as new tasks arrive continuously. Each arrival of a new task corresponds to one epoch of training. The blue line represents the trend of ASO-LoRA's training loss, while the red line corresponds to that of O-LoRA.

### E.3 THE EFFECT ON CATASTROPHIC FORGETTING AND KNOWLEDGE TRANSFER

To further comprehensively evaluate Soft Orthogonality's efficacy in mitigating catastrophic forgetting and facilitating knowledge transfer, we demonstrate the FM and FWT results of ASO-LoRA and O-LoRA on longer benchmarks.

Figure 10 reveals that ASO-LoRA more effectively leverages knowledge from previous tasks even in more complex continual learning scenarios, achieving significantly superior knowledge transfer over O-LoRA across all stages. These results confirm the advantage of our method in knowledge transfer.

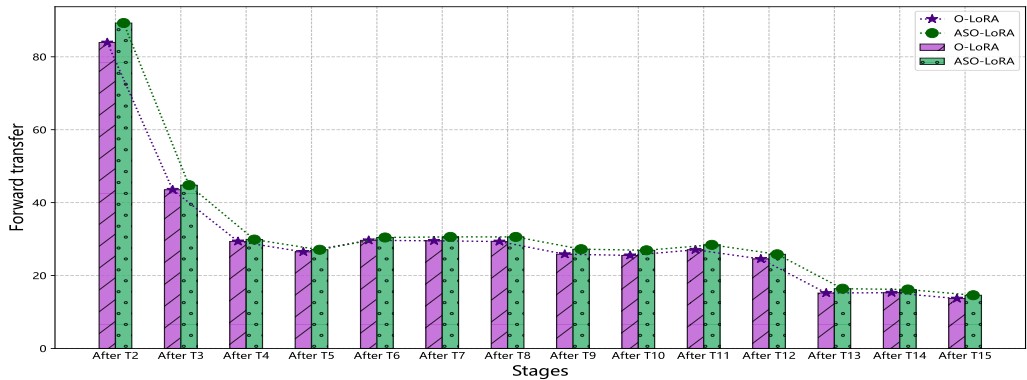

Figure 10: ASO-LoRA vs. O-LoRA on Order4 using metrics FWT. Higher FWT values demonstrate more effective utilization of prior knowledge and superior knowledge transfer.

As illustrated in Figure 11, as the number of tasks increases, ASO-LoRA outperforms O-LoRA in mitigating knowledge forgetting while preserving acquired knowledge across most stages. However, in the final two stages, O-LoRA exhibits less knowledge forgetting than ASO-LoRA. These findings indicate that striking an optimal balance between knowledge transfer and knowledge retention remains an issue worthy of further investigation.

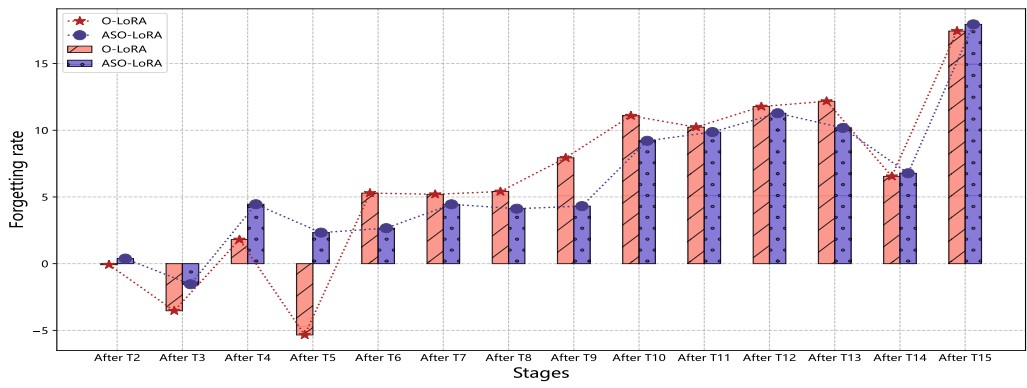

Figure 11: ASO-LoRA vs. O-LoRA on Order4 using metrics FM. Lower FM values indicate stronger resistance to catastrophic forgetting and better knowledge retention capabilities.

## F  LIMITATIONS

While ASO-LoRA has demonstrated strong capabilities in continual learning scenarios through empirical evaluation, several limitations warrant discussion: 1) The observed performance variation across different Transformer architectures, raising an open question regarding whether encoder-decoder frameworks inherently facilitate better knowledge storage. These phenomena suggest underlying mechanistic differences, requiring further investigation. 2) The potential negative impacts

of task overlap need to be further explored. 3) The performance degradation observed in longer benchmarks remains a significant challenge for scaling to more complex real-world applications, such as hundreds of tasks.

We aim to address these limitations in future work to further enhance our method's performance in continual learning scenarios.

