# OpenReview forum: "ASO-LoRA: Attribution Scores-based Soft Orthogonality Low-Rank Adaptation for Large Language Model Continual Learning"
_ICLR.cc/2026/Conference — ICLR 2026 Conference Withdrawn Submission_

### Official Review · Reviewer_HTa5 · 2025-10-29

**Soundness:** 2
**Presentation:** 1
**Contribution:** 1
**Rating:** 2
**Confidence:** 3

**Summary:**

The authors propose **Attribution Scores-based Soft Orthogonality Low-Rank Adaptation (ASO-LoRA)**, a method for continual learning. In a multi-LoRA adapter setting, ASO-LoRA utilizes attribution scores to evaluate task similarity. These scores are based on knowledge neurons: when a specific neuron contributes strongly to task-specific knowledge, its attribution score is high. Task similarity is then expressed through the similarity between their attribution score vectors. Gradient steps are subsequently taken in a softly orthogonal direction relative to task-specific subspaces.

The method is evaluated on short- and long-sequence continual learning benchmarks across T5-Large and LLaMA-2-7B. Additionally, the authors analyze knowledge transfer, forgetting, and output similarity.

**Strengths:**

The paper explores an idea based on attribution-guided orthogonality which is a relevant direction.

---

- The paper includes a nice and diverse set of baselines.

**Weaknesses:**

### **1. Writing Quality and Clarity**

- Numerous typos, formatting issues, and grammatical mistakes.
- Overall poor writing quality makes the paper difficult to evaluate.
- The methodology section is poorly structured and hard to follow.
- The introduction mixes LoRA and multi-LoRA, creating confusion.
- “continual” vs. “continuous” learning

---

### **2. Missing or Insufficient Definitions**

- “Multi-LoRA” is introduced without explanation.
- It is unclear what a knowledge neuron is.
- FM and FWT are not explained nor is interpretation guidance provided.

---

### **3. Methodological Concerns**

- The method essentially resembles adjusted O-LoRA.
- The method requires storing many adapters, raising scalability concerns.
- The proposed method sometimes outperforms what should be an upper bound, unclear why.
- Comparison methods are not actually run, breaking comparability.

---

### **4. Related Work Weaknesses**

- Related work section includes irrelevant content, it resembles more a background section.
- It is not well connected back to the proposed method.

---

### **5. Figures and Presentation Issues**

- Several plots appear stretched with distorted aspect ratios.
- Figure labels and legend text are too small to read. They are sometimes missing or incomplete.
- Figure 3 is unclear about what “current task” means.
- Figure 2’s attribution score is not explained in the description of the figure.

**Questions:**

1. Why does the forward transfer reduce with tasks?

---

### Official Review · Reviewer_bvfs · 2025-10-31

**Soundness:** 2
**Presentation:** 2
**Contribution:** 2
**Rating:** 2
**Confidence:** 5

**Summary:**

This paper proposes a LoRA-based continual learning method for LLMs, ASO-LoRA, which addresses the trade-off between forgetting mitigation as well as knowledge transfer. It computes attribution scores to evaluate task similarity during training and uses the scores as the loss regularization term. The mechanism makes the new task LoRA learning in the soft orthogonal directions between task-specific subspaces, balancing the knowledge transfer and preservation. Experimental results show that ASO-LoRA outperforms other strong baselines.

**Strengths:**

1. ASO-LoRA considers both mitigating forgetting and knowledge transfer, different from some existing continual learning, which only focuses on mitigating forgetting.

2. The paper uses two different kinds of models to conduct experiments.

**Weaknesses:**

1. It’s not clear for computing attribution scores in which training phase. From the statement in line 249, does it mean the interval of computing attribution scores during training is m=20? If so, the computation of ASO-LoRA is very high during training. ASO-LoRA is not efficient for LLMs.

2. The motivation of the proposed method, ASO-LoRA, is not well explored, and some statements lack supporting analysis. For example, in line 180, “We hypothesize that the relationship between LoRA blocks is not strictly orthogonal”, which lacks the reason for discussing this. Also, in line 183, it lacks an empirical or mathematical explanation for why the dynamic soft orthogonality coefficient decreases as more knowledge is shared between LoRAs.”


3. Experiments only utilize classification tasks and single metric average accuracy, which are simple for llama models. For example, in the cited paper, SAPT utilizes SuperNI benchmark, including generation tasks, to evaluate the performance. Furthermore, the ablation study is not sufficient, for example, scaling across different ranks for LoRA, scaling across different learning rates, scaling across different $\lambda$ in the loss function, scaling across different approximation steps, etc. The paper does not show how the performance of ASO-LoRA varies in different settings.

4. ASO-LoRA is not memory-efficient since it linearly grows with the number of sequential tasks.

5. The related works section is too short and lacks analysis.

6. Mathematical expressions in the paper lack consistency, for example, Eq (4) and Eq.(4) are both used.

**Questions:**

1. Why does ASO-LoRA regard A as the core of the task-related subspace and regard B as the linear weighting coefficient of A? Is there any theoretical analysis or strong explanation to support this statement?

2. Why does ASO-LoRA fix the approximation steps at 20? Why does ASO-LoRA not show the performance when choosing different steps? Can authors show the results of scaling across different approximation steps?

3. How to distinguish ASO-LoRA and O-LoRA? Since O-LoRA also utilizes $\lambda$ to control the orthogonality between A subspaces, which could be regarded as soft orthogonality. Does this $\lambda$ not represent the degree of orthogonality? Can authors explain Figure 3 more clearly? It seems like the average range of attribution scores is from 0.1 to 0.3. What if fixing the attribution scores in this range, like 0.2, when computing training loss? It may reduce most computation during training.

4. In Eq (13), what’s the definition of the “Sim” function? What’s the mathematical expression?

---

### Official Review · Reviewer_46in · 2025-11-01

**Soundness:** 3
**Presentation:** 2
**Contribution:** 3
**Rating:** 4
**Confidence:** 4

**Summary:**

This paper proposes ASO-LoRA (Attribution Scores-based Soft Orthogonality Low-Rank Adaptation), a parameter-efficient framework built on multi-LoRA modules. ASO-LoRA first assigns task-specific low-rank subspaces (via separate LoRA blocks) to new tasks, keeping the LLM’s base weights frozen for efficiency.

**Strengths:**

1. ASO-LoRA addresses a key limitation of hard orthogonality methods (e.g., O-LoRA) by using attribution scores to dynamically adjust subspace overlap. This enables intentional knowledge transfer between similar tasks.

2. As a plug-in framework, ASO-LoRA adds no extra trainable parameters beyond multi-LoRA modules.

**Weaknesses:**

1. The paper uses integrated gradients (with Riemann approximation) to compute attribution scores for knowledge neurons but provides no justification for critical choices—e.g., why 20 approximation steps (m=20) or how noise in gradient estimates affects score reliability.

2. Key hyperparameters are set without empirical explanation. The paper does not test how λ affects the balance between transfer and forgetting, nor why Spearman’s correlation is preferred over other similarity metrics (e.g., Pearson’s).

3. Only evaluates classification tasks (sentiment, topic, NLI, QA). It does not test complex tasks like open-ended generation (summarization, dialogue) or reasoning (math, logic), where token-level frequency cues may behave differently.

4. Only tested on T5 and LLaMA2. It is unknown if ASO-LoRA works for newer models (e.g., LLaMA3-8B, Qwen3-8B).

5. In some cases, ASO-LoRA is worse than the baseline O-LoRA.

6. The writing can be improved.

**Questions:**

1. How sensitive is ASO-LoRA to the number of Riemann approximation steps (m) for attribution scores? Would increasing m improve score reliability, or introduce unnecessary computational cost?

---

### Official Review · Reviewer_pyRx · 2025-11-06

**Soundness:** 2
**Presentation:** 2
**Contribution:** 2
**Rating:** 2
**Confidence:** 4

**Summary:**

This paper proposes ASO-LoRA, a parameter-efficient continual learning framework for large language models that mitigates catastrophic forgetting while enabling knowledge transfer. By using multiple LoRA modules to assign task-specific subspaces and leveraging attribution scores to guide soft orthogonal gradient updates, ASO-LoRA achieves efficient adaptation without task labels and outperforms existing methods on standard benchmarks.

**Strengths:**

1. The investigated problem of continual learning using low-rank adaptation is important.

2. The experimental evaluation incorporates a wide range of baselines.

**Weaknesses:**

1. The writing quality of the paper requires substantial improvement, as it contains many instances of non-standard usage. For example, not every equation should have a label, and each formula line must be followed by appropriate punctuation. The notation should also follow standard conventions, such as using `\log` instead of *log*, and `\operatorname{Sim}()` instead of *Sim()*.

2. The notation in the paper is not clearly defined. For instance, what does $\psi_{t, \mathcal{T}}^{\text {soft }}$ represent? Is it a scalar or a matrix?

3. The improvements over previous works appear to be minor, as shown in Table 1. Moreover, no standard deviation is provided, and after conducting a t-test, the proposed method may not be significantly superior to others.

**Questions:**

See weaknesses above.

---

### Note · Authors · 2025-11-13

I have read and agree with the venue's withdrawal policy on behalf of myself and my co-authors.